# Chemical Characteristics of Atmospheric PM_10_ and PM_2.5_ at a Rural Site of Lijiang City, China

**DOI:** 10.3390/ijerph17249553

**Published:** 2020-12-20

**Authors:** Yu Liu, Xurui Li, Wan Wang, Baohui Yin, Yuanguan Gao, Xiaoyang Yang

**Affiliations:** 1School of Pharmaceutical Sciences, Liaoning University, Shenyang 110036, China; liuyu@lnu.edu.cn (Y.L.); 4031932273@smail.lnu.edu.cn (X.L.); 2State Key Laboratory of Environmental Criteria and Risk Assessment, Chinese Research Academy of Environmental Sciences, Beijing 100012, China; yinbh@craes.org.cn (B.Y.); gaoyg@craes.org.cn (Y.G.); yangxy@craes.org.cn (X.Y.)

**Keywords:** PM_10_ and PM_2.5_, chemical components, PAHs and NPAHs, biomass burning, long range transport

## Abstract

Emissions from biomass burning are very serious in Southeast Asia and South Asia in April. In order to explore the effect of long-range transport of biomass emissions from the Indochina Peninsula in Southwest China during the period of the southeast monsoon season and to find out the main pollution sources in local atmospheric PM_2.5_, a field campaign was conducted from 6–26 April 2011 in Lijiang, China. Twenty-four-hour PM_10_ and PM_2.5_ filter samples were collected, and inorganic ions, elements, and carbonaceous components (including organic carbon (OC) and elemental carbon (EC), polycyclic aromatic hydrocarbons (PAHs) and nitrated PAHs (NPAHs)) were measured. The monthly mean mass concentrations of particulate matter (PM) were 40.4 and 14.4 μg/m^3^ for PM_10_ and PM_2.5_, respectively. The monthly mean concentrations of OC and EC in PM_10_ were 6.2 and 1.6 μg/m^3^, respectively. The weekly mean concentrations of ∑PAHs and ∑NPAHs were 11.9 ng/m^3^ and 289 pg/m^3^, respectively, in atmospheric PM_10_ of Lijiang. The diagnostic ratios of PAH and NPAH isomers were used to analyze the sources of PAHs and NPAHs in PM_10_. The ratios of Benz(a)anthracene/(Chrysene+Benz(a)anthracen), Fluoranthene/(Fluoranthene+Pyrene) and Indeno(1,2,3-cd)pyrene/(Benzo(g,h,i)perylene+Indeno(1,2,3-cd)pyrene) were 0.45 ± 0.04, 0.61 ± 0.01, and 0.53 ± 0.03, respectively, indicating the contribution from coal combustion and biomass burning. The 1-nitropyrene/Pyrene (1-NP/Pyr) ratio was 0.004 ± 0.001, suggesting that the contribution to NPAHs mainly came from coal combustion. Sulfate was the most prominent inorganic ionic species, with monthly mean levels of 2.28 and 1.39 μg/m^3^ in PM_10_ and PM_2.5_, respectively. The monthly mean mass ratios of NO_3_^−^/SO_4_^2−^ were 0.40 and 0.23 in PM_10_ and PM_2.5_, respectively, indicating that the contribution of atmospheric anions from coal combustion sources was much more important than that from other sources. Based on the relatively high SO_4_^2−^ concentrations and low NO_3_^−^/SO_4_^2−^ ratios, combined with the data analysis of isomer ratios of PAHs and NPAHs, we can conclude that coal combustion, traffic, and dust were the major contributors to local atmospheric PM in Lijiang city, while biomass burning may also have contributed to local atmospheric PM in Lijiang city to some degree.

## 1. Introduction

Many gases and aerosols emitted from biomass combustion have an important impact on the global climate system, the atmospheric environment, and ecosystems [1,2,3]. Biomass burning in the Indochina Peninsula, Southeast Asia, has long been associated with certain vegetation types and land uses there [4], and springtime emissions from biomass burning have been linked to significant regional impacts, which led to the area being considered a global hotspot in terms of poor air quality [5,6]. Under the influence of prevailing westerly winds, pollutants emitted from biomass burning in the Indochina Peninsula can be transported to Southern China and the Pacific Northwest, affecting air quality where the polluted air mass passes by. Especially in springtime (so-called dry season or pre-monsoon), some ground observations in Southern China have shown that the concentrations of PM_2.5_, PM_10_, CO, black carbon (BC), ozone, and brown carbon increase significantly when the southwest wind from the Indochina Peninsula blows in the spring [7,8,9,10].

Moreover, biomass burning and transport from Indochina can also affect the radiation characteristics of aerosols in Southwest China. Biomass combustion smoke in clouds can increase the net absorption of atmospheric radiation [11]. Through numerical simulation, it was found that there are two main paths for biomass burning in Southeast Asia to impact China; one path is from Myanmar to Yunnan and other places, and the monthly mean contribution to Yunnan (PM_2.5_) reaches 20 μg/m^3^ (a contribution of 30%, twice as much as from local biomass combustion), and the daily contribution can even reach 34 μg/m^3^ (with a contribution of 43%) [12].

Numerous studies have shown that long-range transport of biomass combustion emissions not only has an impact on the atmospheric composition of the carrier gas mass [10,13,14], but also heavily polluted gas masses carry a large amount of particulate matter to areas where they are transported [9]. Biomass burning in the Indochina Peninsula region is an important source of atmospheric mercury (Hg), and total mercury emissions from Indochina are estimated to be 28.5 Mg/yr from 2001 to 2008, and 40% of that total is from biomass burning [15].

Air pollutant monitoring at relatively clean sites can be used to study the pollution characteristics of different air masses and the main sources of highly polluted air masses and to explore the impact of long-range transport on regional atmospheric pollution [16,17,18]. Some studies have shown that the main sources of atmospheric volatile organic compounds in background sites mainly came from masses of long-range transport and natural sources [16]. In addition, the fine particle pollution in high mountain areas mainly came from nearby urban areas by regional transport [17]. It was also reported that long-range transport of polycyclic aromatic hydrocarbons (PAHs) and nitrated polycyclic aromatic hydrocarbons (NPAHs) at a rural site in New Zealand during winter was from nearby countries [18].

It is an extremely important scientific issue to study the possible impact of biomass burning emissions transported from Southeast and South Asia on the atmospheric environment in China. The observation site in this study, located in Southwest China on the transmission channel of the spring monsoon, was relatively clean. Thus, it is possible to understand the local atmospheric conditions and the impact of polluted air masses such as biomass burning emissions carried by the spring monsoon by studying the physicochemical characteristics of local atmospheric fine particulate matter. In addition, the research results based on this site can help improve China’s understanding of the long-range transport of air pollutants from South and Southeast Asia and can help provide a scientific basis for medium- and long-term decision-making on regional air pollution prevention and control.

## 2. Materials and Methodology

### 2.1. Sampling Site

Lijiang City (100.25° E, 26.86° N) is located in Yunnan Province, Southwest China. It is 500 km south of Kunming City (the provincial capital). The geographical location is shown in Figure 1. Lijiang, a well-known international tourist city, sits at an altitude more than 2400 m above sea level. The sampling site is located in the southwest suburb of the city, which is less polluted by human-generated emissions and can be regarded as a clean background site. Aerosol sampling instruments were set on the roof of the Lijiang Environmental Protection Bureau building, 10 m above the ground.

### 2.2. Aerosol Sampling 

Both PM_10_ and PM_2.5_ were intensively collected by medium flow samplers (Dickel-80 model, Beijing Geological Instrument-Dickel Cooperation Limited, Beijing, China) with a flow rate of 78 L/min. Quartz membrane filters (2500QAT-UP) produced by the Pall Corporation (USA) were used to collect atmospheric particles. Before sampling, the filter membranes were placed in a muffle furnace at 600 °C for 2 h to remove background organic matter.

Biomass burning emissions in Southeast Asia and South Asia are rather severe from April to May every year, which is the pre-monsoon season. The continuous field campaign lasted for 21 days, from 6–26 April 2011. PM_10_ and PM_2.5_ were collected over 23 h, from 10:00 to 09:00 the next day, from 7–26 April. In addition, extra PM_10_ membrane filters were collected from 6–13 April using high-volume samplers (TH-1000H model, Wuhan Tianhong Environmental Protection Industry Cooperation Limited, Wuhan, China) with a flow rate of 1.0 m^3^/min, which were to be used for PAH and NPAH analysis.

### 2.3. Analytical Method

PM mass concentrations were gravimetrically determined with a micro-balance before and after sampling. The sample filters were set out for 24 h at constant room temperature (20 ± 1 °C) and relative humidity of approximately 40% and then weighed with an electronic balance (Shimadzu Instruments Co., Ltd., Kyoto, Japan).

Organic carbon (OC), elemental carbon (EC), and total carbon were determined via thermal/optical reflectance (TOR) [19]. PAH and NPAH analysis was conducted using high-performance liquid chromatography (HPLC), and fluorescence and chemiluminescence detectors were deployed for PAHs and NPAHs, respectively, as described in detail by Yang et al. [20]. All the compounds of PAHs and NPAHs were listed in Table 1. A certain area of filter was extracted via quantitative deionized water ultrasonic vibration, and the extract solution was filtered using 0.47 mm microfilm. The cations and anions in extract solution, including Na^+^, NH_4_^+^, K^+^, Ca^2+^, Mg^2+^, F^−^, Cl^−^, NO_3_^−^, and SO_4_^2−^, were analyzed with a DX500 ionic chromatography (IC) instrument made by Dionex (Sunnyvale, CA, USA), as described by Wang et al. [21]. Inductively coupled plasma mass spectrometry (ICP-MS) was performed on a NexION 300X (PerkinElmer, Walsham, MA, USA) to analyze 25 elements, including Al, Fe, Ca, Na, and K [22]. The uncertainty of PM mass determination and the analysis of carbonaceous and organic species are estimated at 15% maximum for concentrations well above the detection limit. Normally, the uncertainty of other species is less than 10%.

## 3. Results and Discussion

### 3.1. PM_10_ and PM_2.5_ Concentrations

During the monitoring period in Lijiang in April 2011, monthly mean concentration of PM_10_ and PM_2.5_ was 40.4 and 14.4 μg/m^3^, respectively. The daily PM_10_ concentration ranged from 21 to 76 μg/m^3^. The daily PM data are shown in Figure 2. The PM_10_ concentration exceeded the daily mean limit of the national first standard (50 μg/m^3^) on 5 days [23]. The daily PM_2.5_ concentration ranged from 8 to 23 μg/m^3^ and did not exceed the daily mean limit of national first standard (35 μg/m^3^) throughout the monitored period [23].

Some studies have focused on the clean background area, and it was reported that the annual mean concentration of PM_10_ and PM_2.5_ at four national atmospheric background (NAB) sites from northern to southern China was 29 and 17 μg/m^3^, respectively [24]. Among these four sites (Table 2), annual mean PM_10_ ranged from 18–44 μg/m^3^ and annual mean PM_2.5_ ranged from 13–21 μg/m^3^ in 2013 [25].

The PM_10_ and PM_2.5_ concentrations in Lijiang were compared with the PM monitoring data of some foreign rural sites, as shown in Table 1, in which PM_10_ and PM_2.5_ concentrations are all annual mean values, except three seasonal values. The Gosan super site is located in Jeju Island, South Korea, and the annual mean values of local atmospheric PM_10_ and PM_2.5_ were 28.4 and 17.2μg/m^3^, respectively [26]. Tokchok, an island of South Korea, had an annual mean PM_2.5_ of 18.7 μg/m^3^ [27]. K-puszta is a continental (rural) background air monitoring station in Hungary, operated within the framework of the European Monitoring and Evaluation Program (EMEP). At this site, the PM_10_ and PM_2_ seasonal mean concentrations were, respectively, 24 and 13.6 μg/m^3^ in the warm and dry periods of summer 2003 [21] and 25 and 17.4 μg/m^3^ in the warm period of 2006 [28,29]. The trends and variability of PM_10_ and PM_2.5_ concentrations at four rural background sites in five European countries for the period 1998 to 2010 were investigated, and the annual mean PM_10_ and PM_2.5_ values ranged from 12–25 and 8–20 μg/m^3^, respectively [30]. The monthly mean PM_10_ concentration in Lijiang was higher than the values in South Korea and Europe, but the monthly mean PM_2.5_ concentration was comparable with the monitoring values abroad.

During this observation period, the monthly mean ratio of PM_2.5_/PM_10_ was 0.37, with daily values ranging from 0.20 to 0.58 in Lijiang. The inhalable particles at the Lijiang site were mainly coarse, which indicates that the proportion of fine particles from various anthropogenic emission sources such as coal fire, local motor vehicles, and industrial sources was relatively low, while the contribution from dust sources cannot be ignored. The seasonal mean fraction of fine-sized PM_10_ (PM_2.0_ for 2003 and PM_2.5_ for 2006) was 0.57 in 2003 and 0.67 in 2006 in K-puszta, Hungary [21,28,29]. By comparison, the monthly mean ratio of PM_2.5_/PM_10_ in Lijiang was consistent with the four NAB sites in China [24], but lower than in K-puszta [21,28,29].

### 3.2. OC/EC 

As shown in Figure 3, the monthly mean concentrations of OC and EC in atmospheric PM_10_ in Lijiang were 6.2 and 1.6 μg/m^3^ and their daily concentration ranges were 2.4–12.1 and 1.1–2.3 μg/m^3^, respectively. The monthly mean concentrations of OC and EC in atmospheric PM_2.5_ in Lijiang were 3.6 and 1.2 μg/m^3^ and their daily concentration ranges were 1.5–7.9 and 0.7–2.0 μg/m^3^, respectively. Obviously, most carbonaceous particles were in fine size fraction, especially for EC. The fraction of OC in PM_10_ and PM_2.5_ was 16 and 25%, respectively; the fraction of EC in PM_10_ and PM_2.5_ was 4.3 and 8.7%, respectively. Correlation analysis between OC and EC was examined with the SPSS software (IBM, Armonk, NY, USA). OC and EC concentrations showed a high correlation (with a correlation coefficient of 0.84) in PM_2.5_ but not PM_10_ in Lijiang, which may mean that OC and EC in PM_2.5_ have the same emission sources.

Carbonaceous data of other rural sites were sorted, and the monthly mean concentration of OC in atmospheric PM_2.5_ of Lijiang (4.9 μg/m^3^) was comparable to the values obtained at the Jianfengling site (3.1 μg/m^3^) in Hainan Province [31], the Hezui site (4.9 μg/m^3^) in Hong Kong [31], and the Dinghushan site (5.1 μg/m^3^) in Zhejiang Province [32]. The monthly mean concentration of OC in PM_2.5_ at the Lijiang site was three times lower than that obtained at the Tengchong site (16.8 μg/m^3^) in Yunnan Province [31] but one order of magnitude higher than that at the Nam Co site (0.09 μg/m^3^) [33]. At the Gosan site in South Korea, the OC and EC concentrations in atmospheric PM_2.5_ were 4.0 and 1.7 μg/m^3^ [26], respectively, slightly higher than those at Lijiang site.

The OC/EC ratio can be used to identify the emission and conversion characteristics of carbon aerosols and to evaluate and identify the secondary sources of particulate matter [34]. It is considered that the secondary reaction exists when the OC/EC ratio is higher than 2 [35]. A relatively high OC/EC ratio indicates that OC may partly come from the photochemical reaction of secondary organic particles. In addition, biomass burning releases more OC particles, which can also lead to a high OC/EC ratio [36]. The OC/EC ratios from different combustion sources were studied. The averaged OC/EC ratio of biomass burning in China was 7.7, and the main cereal straw includes rice, wheat, and corn [37]. The OC/EC ratio of diesel vehicles ranges between 0.92 and 2.5 [38] and between 0.3 and 7.6 for coal combustion sources [36]. The OC/EC ratios in atmospheric PM_10_ and PM_2.5_ of Lijiang were 3.8 and 3.1, respectively; the OC/EC ratios in PM_2.5_ ranged widely from 1.1 to 5.4. The OC/EC ratio of Lijiang indicates that the long-range transport of biomass burning air mass and coal combustion sources seems to contribute a certain amount to the carbonaceous particles in Lijiang. However, far away from the burning field, the OC particles carried by the air mass would age during long-range transport. Therefore, the OC/EC ratio cannot explain to what extent PM_10_ and PM_2.5_ in Lijiang have been affected by local combustion or transport from Southeast Asia.

### 3.3. PAHs and NPAHs

The individual and weekly mean concentrations of 10 PAH isomers in atmospheric PM_10_ of Lijiang are shown in Table 3. The weekly mean concentration of detected ∑PAHs was 11.9 ng/m^3^, which is much lower than the values obtained at urban sites. For example, the seasonal mean concentration of ∑PAHs was 191 ng/m^3^ in PM_2.5_ of Lanzhou in winter [39], 10 times that of Lijiang. In the atmospheric PM_10_ of Yuxi City (Yunnan Province), the annual mean concentration of ∑PAHs, 11.7 ng/m^3^, was very close to that of Lijiang [40]. Similar studies of PAHs were conducted at four NAB sites in China. Their daily mean concentrations of ∑PAHs in PM_10_ and PM_2.5_ were in the range of 0.13–30 ng/m^3^ and 0.09–25 ng/m^3^ [41], respectively, which are comparable to the daily values of Lijiang. Pangquangou is one of the four sites, which is located in Shanxi Province and is most seriously affected by coal combustion. In the spring of 2013, the seasonal mean concentration of ∑PAHs in PM_10_ at Pangquangou was 13 ng/m^3^ [41], which is slightly higher than that of Lijiang. The daily variation of ∑PAHs and each isomer concentration in PM_10_ of Lijiang was 3 to 4 times and was not as large as that of the four NAB sites.

Fluoranthene (Flt) accounted for the highest proportion (22%) of ∑PAHs, with a weekly mean concentration of 2.7 ± 1.4 ng/m^3^ in atmospheric PM_10_ of Lijiang. The monthly mean PAH concentration of Flt was found at night (0.06 ± 0.13 ng/m^3^), with the nocturnal range of ND (not detected) to 0.14 ng/m^3^ [42]. The highest PAH content was Flt in atmospheric PM_2.5_ of Mt. Wuzhishan (a NAB site in Hainan Province, South China), with an annual mean concentration of 0.42 ng/m^3^ [43].

Benzo(a)pyrene (BaP) has high toxicity and showed a weekly mean concentration of 1.1 ng/m^3^, with the daily concentration ranging between 0.6 and 2.0 ng/m^3^, in atmospheric PM_10_ of Lijiang, and each daily BaP concentration was obviously lower than the daily mean limit of the national standard (2.5 ng/m^3^) [23]. The weekly mean concentration of BaP in atmospheric PM_10_ of Lijiang was much lower than that in PM_10_ (11.8 ng/m^3^) [39] at Lanzhou, an urban site, in winter 2012, and was slightly higher than that in PM_10_ (0.73 ng/m^3^) [39] at Lanzhou in summer 2013 and in PM_2.5_ (0.7 ng/m^3^) [41] at a rural site (Pangquangou) in spring 2013., but was significantly higher than that in PM_2.5_ (0.027 ng/m^3^) [43] at Mt. Wuzhishan. There was a significant linear correlation between BaP and ∑PAH concentrations, with a correlation coefficient of 0.9878. Based on the data comparison and analysis of PAH concentrations, we can conclude that PAH pollution in atmospheric PM_10_ of Lijiang in spring 2011. was not very serious, but more attention still needs to be paid to long-term and continuous monitoring of PAHs in order to explore the main sources of atmospheric pollution at this rural site.

There are many kinds of PAH isomers, which are greatly affected by different combustion types and conditions [44]. The common method for source apportionment of PAHs is the use of ratios, which is simple and widely used [20,39,45,46,47,48,49,50,51,52,53]. Although the source apportionment of PAHs based on PAH isomer ratios may have certain limitations [53], the ranges of characteristic ratios for biomass (straw, wood, and shrub combustion, etc.), fuel oil, and coal combustion are different, which can provide a certain useful judgment basis. The diagnostic ratios of PAH isomers used to estimate the source characteristics of PAHs in this study are listed in Table 4. Benz(a)anthracene/(Chrysene+Benz(a)anthracen), Fluoranthene/(Fluoranthene+Pyrene) and Indeno(1,2,3-cd)pyrene/(Benzo(g,h,i)perylene+Indeno(1,2,3-cd)pyrene) will be presented by using their abbreviation IDP/(BghiPe + IDP), BaA/(Chr + BaA) and Flt/(Flt + Pyr) for the later discussion. 

The daily concentrations of PAH isomers at Lijiang varied moderately (see Table 3), but three diagnostic ratios used to estimate PAH sources were quite constant (see Table 4). This may indicate that the source or formation process of PAHs at Lijiang was stable. In this study, the weekly mean ratio of IDP/(BghiPe+IDP) was 0.53 ± 0.04, which is in the ranges of biomass burning, coal combustion, and diesel vehicles. Therefore, these three sources might contribute to the PAHs in atmospheric PM_10_ of Lijiang. The weekly mean ratio of BaA/(Chr + BaA) was 0.45 ± 0.01, which is in the ranges of coal combustion but a little bit lower than that of biomass burning. Therefore, these two sources may contribute to the PAHs of atmospheric PM_10_ of Lijiang. The weekly mean ratio of Flt/(Flt + Pyr) was 0.61 ± 0.03, which is in the ranges of diesel vehicles but a little bit higher than that of biomass burning. In this study, the Flt/(Flt + Pyr) ratio might be affected by the emissions of biomass burning, coal combination, and diesel vehicles together. Based on the comprehensive comparison and analysis of PAH data of Lijiang, it looks like the PAHs in atmospheric PM_10_ of Lijiang are affected by coal combustion, vehicle emission, and biomass burning sources. However, when considering the sampling site is located in a rural site, traffic emissions were not important at that time. Thus, coal combustion and biomass burning could be the major sources for atmospheric PAHs in Lijiang.

NPAHs are derivatives of PAHs and are formed by substituted nitro groups, which have stronger mutagenic, carcinogenic, and teratogenic toxicity than the parent PAHs. In this study, the weekly mean concentration of 16 kinds of ∑NPAHs was 289 pg/m^3^ (see Table 5). Among them, the concentration of 2-NTP (weekly mean of 230 pg/m^3^) was the highest, followed by 9-nitroanthracene (9-NA) (66 pg/m^3^) and 6-NC (22 pg/m^3^). Similar studies on NPAHs have also been done in village fields and rural sites elsewhere. For example, in Chiang Mai and several other provinces in northern Thailand, 9-NA (249 pg/m^3^) was the most abundant NPAH, which suggests that it is generated from biomass burning during the dry season [54]. In a sugarcane burning region, the highest average concentrations were obtained for 9-NA among the NPAH compounds in diurnal and nocturnal samples (1.5 ± 1.2 and 1.3 ± 2.1 ng/m3, respectively) [42]. In both urban and rural areas of northern China, among 12 detected isomers, 9-NA was the most abundant NPAH, with daily concentrations in a wide range of 38–694 pg/m^3^ in rural fields [55]. NPAHs were studied at a rural site, and the research results showed that wood burning at low temperatures tends to produce low ring number NPAHs [56]. The NPAH isomers in atmospheric PM_10_ of Lijiang are mainly of low ring number (2–3 rings), which is different from developed countries, where atmospheric NPAHs are dominated by high ring numbers (4 rings), such as in Tokyo, Japan [18]. It seems that 9-NA can be used as a good tracer for biomass burning, and 9-NA has the second highest content of NPAHs in PM_10_ in Lijiang. Thus, in conclusion, biomass burning must have had some influence on local atmospheric NPAHs of Lijiang in April 2011. Considering the effect of long-range transport of Southeast Asian air masses to Southwest China in Spring [10,12], probably, the atmospheric pollutants from biomass burning were also carried by air masses along with the monsoon to China in spring during the period of our observation.

Moreover, many ratios of NPAH isomers have been used to trace their sources. The 1-NP/Pyr ratio is usually used as an important indicator for source apportionment of PAH and NPAH, with values of 0.36 for vehicle exhaust and 0.001 for coal combustion [20]. In this study, the calculated weekly mean ratio was 0.004 ± 0.001, indicating the importance of contribution from coal combustion.

It is proposed that the 9-Nitroanthracene/1-Nitropyrene(9-NA/1-NP) ratio can be regarded as a new indicator for assessing the contribution of biomass burning, and a monthly mean 9-NA/1-NP ratio higher than 10 indicates wood combustion as the source and less than 10 indicates motor vehicle exhaust emissions as the source [54]. In this study, the weekly mean 9-NA/1-NP ratio was around 6.1. Therefore, it is estimated that coal combustion and vehicle exhaust could contribute a lot to PAHs and NPAHs of atmospheric PM_10_ in Lijiang, and maybe biomass burning contributes a little as well.

### 3.4. Water-Soluble Inorganic Ions

In this study, the monthly mean concentrations of total ions in atmospheric PM_10_ and PM_2.5_ were 5.2 and 2.8 μg/m^3^, accounting for 13 and 23%, respectively. SO_4_^2−^, NO_3_^−^, Ca^2+^, and NH_4_^+^ were the main ions, as shown in Figure 4, and their monthly mean concentrations were 2.4, 0.86, 1.08, and 0.30 μg/m^3^ in PM_10_ and 1.6, 0.35, 0.30, and 0.26 μg/m^3^ in PM_2.5_, respectively. According to previous studies at rural sites, the seasonal percentage range of water-soluble ions in PM_2.5_ (35.5–42.2%) at four NAB sites in China was higher than that in PM_10_ (25.7–33.3%) [25]. The monthly mean percentage of water-soluble ions in PM_2.5_ in Lijiang was 23 ± 9%, ranging daily from 11 to 43%.

In this study, Ca^2+^ was the dominant cation in PM_10_ and PM_2.5_, with the monthly mean mass percentages of 60 and 34% in total cations, respectively, which was much higher than those of NH_4_^+^. This indicates that the cations in atmospheric inhalable particles of Lijiang were mainly from dust, instead of anthropogenic sources. SO_4_^2−^ was the dominant anion in PM_10_ and PM_2.5_, with monthly mean mass percentages of 70 and 77%, respectively. These percentages were much higher than those of NO_3_^−^ (25% of both PM_10_ and PM_2.5_) in total anions.

In this study, the monthly mean mass percentages of major ions (SO_4_^2−^, NO_3_^−^, and NH_4_^+^) accounted for 72 and 67% of total ions and 9.2 and 17% of PM_10_ and PM_2.5_, respectively. Compared with the major ionic concentrations at domestic and international rural sites (Figure 5), the major ionic concentrations in atmospheric PM_2.5_ of Lijiang were at rather lower levels, only slightly higher than those at the Norikura site in Japan [57], but lower than those at other sites [26,27,28,58,59,60,61].

The daily ranges of NO_3_^−^/SO_4_^2^ mass ratios in atmospheric PM_10_ and PM_2.5_ at the four NAB sites were 0.11–0.70 and 0.09–0.41, respectively [25]. The monthly mean NO_3_^−^/SO_4_^2^ mass ratios in atmospheric PM_10_ and PM_2.5_ of Lijiang were 0.40 and 0.23, respectively, which was comparable to those of background sites. Apparently, the NO_3_^−^/SO_4_^2−^ mass ratios were significantly less than 1 in Lijiang, which means that SO_4_^2–^ was the predominant anion. This may indicate that atmospheric PM_10_ and PM_2.5_ of Lijiang are more affected by emissions from coal combustion than from vehicle exhausts.

In this study, the monthly mean concentrations of K^+^ in PM_10_ and PM_2.5_ of Lijiang were 0.15 and 0.06 μg/m^3^, respectively, and the monthly mean PM_10_ fraction of K^+^ in the fine size of PM_2.5_ was 0.42, which indicates that the K^+^ in atmospheric PM_10_ of Lijiang was mainly from coarse fraction and probably from crustal sources. The daily concentration ranges of K^+^ in PM_10_ and PM_2.5_ were 0.11–0.31 and 0.10–0.26 μg/m^3^, respectively, at the four NAB sites. The monthly mean K^+^ concentration in PM_10_ of Lijiang was comparable to the values at the four NAB sites, while K^+^ concentration in PM_2.5_ of Lijiang was significantly lower than the values at those sites [25]. Water-soluble K^+^ of the local atmosphere is a good tracer for biomass burning in the near-field of combustion [62]. However, in cities, the sources of K^+^ are diverse, such as combustion from coal and fuel oil, and the content of K^+^ from biomass burning is not dominant [63]. In Lijiang, the proportion of K^+^ in PM_2.5_ was around 0.4%, much less than the 2.5% in the near-source biomass burning in northern Indochina [64]. Thus, only the K^+^ concentrations in PM_2.5_ of Lijiang cannot explain whether the local atmosphere is affected by biomass burning of long-range transport from Southeast Asia, or the extent of the impact.

### 3.5. Elemental Composition

In this study, the order of monthly mean elemental concentration was as follows: Ca, S, Mg, Fe, Al, Na, K, and Zn. The monthly mean concentrations of main elements are shown in Table 6. The elemental concentrations of atmospheric PM_2.5_ in rural sites of different regions in China, including Kunming city (provincial capital of Yunnan), near Lijiang city, are also shown in Table 6. In the table, combined data of the four NAB sites are annual mean values, as well as in Mt. Dinghu, Kunming, and Beijing, and the monthly mean value in Xinglong in September.

In this study, the monthly mean concentration of Ca (a crustal element) was the highest among the measured elements. The monthly mean concentrations of Ca in PM_10_ and PM_2.5_ were 2.08 and 0.59 μg/m^3^, respectively. Among the polluting elements, the concentration of Zn was the highest, with monthly mean concentrations of 0.022 and 0.017 μg/m^3^ in PM_10_ and PM_2.5_, respectively. The monthly mean concentrations of Cr and Cu ranged from 0.008 to 0.009 μg/m^3^ in PM_10_ and PM_2.5_. The monthly mean concentrations of As in PM_10_ and PM_2.5_ were 0.00023 and 0.00018 μg/m^3^, respectively, which did not exceed the annual mean limit of the national standard (0.006 μg/m^3^) [23].

As shown in Table 5, the monthly mean concentrations of crustal elements (Ca, Mg, Fe, K, and Mn) in PM_2.5_ of Lijiang are comparable to those of clean sites, such as Mt. Dinghu from autumn to winter in 2006 [65] and the four NAB sites in spring 2013 [23], and about five times lower than that at Xinglong in autumn 2008, which is near the megacity, Beijing [66]. The monthly mean concentrations of trace elements (Zn, Cu, Cr, and As) in atmospheric PM_2.5_ of Lijiang are comparable with the values at the four NAB sites in spring 2013 [24], and nearly one order of magnitude lower than those at other sites [65,67,68]. The monthly mean concentrations of polluting elements in PM_2.5_ of Lijiang were low, which indicates that most heavy metals seemed to be less affected by human activities in Lijiang. As an example, the monthly mean concentration of As (0.00018 μg/m^3^) in PM_2.5_ of Lijiang was much lower than that of Kunming City (0.03 μg/m^3^) in spring 2013 [67] and also lower than that of Beijing in spring 2012 (0.012 μg/m^3^) [68].

The enrichment factor (EF) is an important index to distinguish and evaluate whether elements come from anthropogenic or crustal/soil sources, and the calculation formula and usage method are well described elsewhere [69,70,71], and to what degree the given elements in environmental samples are enriched or not from anthropogenic sources can be assessed using their EFs. The EF values of the detected elements in atmospheric PM_10_ and PM_2.5_ of Lijiang were calculated using the crustal element Fe as the reference in this study. The enrichment factor values of elements in PM_2.5_ are slightly higher than those in PM_10_. In PM_2.5_ of Lijiang, the monthly mean enrichment factor of element S was the largest, reaching 492, followed by Zn, Cu, Cr, and As, with values of 111, 59, 29, and 26, respectively. Thus, these five elements are regarded as significantly polluting elements by anthropogenic activities. As is a trace element typically produced by coal combustion [72], and it is confirmed that elevated atmospheric As, Cd, and Cr levels in Yunnan and Guizhou provinces were attributed to the high metal content of local coal resources in Southwest China [73]. Thus, according to the data analysis of metal characteristics, it is deduced that local coal combustion has a certain impact on atmospheric PM_10_ and PM_2.5_ in Lijiang.

## 4. Conclusions

In the present study, the monthly mean mass concentrations of PM_10_ and PM_2.5_ were 40 and 14.4 μg/m^3^, respectively. The monthly mean ratio of PM_2.5_/PM_10_ was 37% (with the daily range of 20–58%), indicating that inhalable particles in Lijiang in spring 2011 were mainly coarse particles. The characteristics of PM concentration were consistent with those of Lijiang City as a rural area.

The monthly mean concentrations of OC and EC in atmospheric PM_10_ of Lijiang were 6.2 and 1.6 μg/m^3^ and in atmospheric PM_2.5_ were 3.6 and 1.2 μg/m^3^, respectively. Obviously, most carbonaceous particles were in fine size fraction, especially for EC.

The weekly mean concentrations of ∑PAHs and ∑NPAHs in atmospheric PM_10_ were 11.9 ng/m^3^ and 289 pg/m^3^, respectively. The ratios of BaA/(Chr + BaA), Flt/(Flt + Pyr), and IDP/(BghiPe + IDP) were 0.45 ± 0.04, 0.61 ± 0.01, and 0.53 ± 0.03, respectively, and the 1-NP/Pyr ratio was 0.004 ± 0.001. A comprehensive comparison and analysis of PAH and NPAH data in atmospheric PM_10_ show that coal combustion can be the major source for atmospheric PAHs and NPAH in Lijiang and more or less from biomass burning and vehicle exhaust may be as well.

The monthly mean concentrations of total ions in atmospheric PM_10_ and PM_2.5_ were 5.2 and 2.8 μg/m^3^, respectively, and SO_4_^2−^, NO_3_^−^, Ca^2+^, and NH_4_^+^ were the main ions. The monthly mean NO_3_^−^/SO_4_^2−^ mass ratios were 0.40 and 0.23 in PM_10_ and PM_2.5_ samples, respectively, indicating that the contribution of major ions transformed from combustion sources was much more important than that from motor vehicle sources.

Among the elements measured, the crustal element Ca presented the highest concentration, with monthly mean values of 2.08 and 0.59 μg/m^3^ in PM_10_ and PM_2.5_, respectively. As for the heavy metals, Zn presented the highest concentration, with monthly mean values of 0.022 and 0.017 μg/m^3^ in PM_10_ and PM_2.5_, respectively. Highly enriched elements (S, Zn, Cu, Cr, and As) have been significantly influenced by anthropogenic pollution, mostly from coal combustion.

Considering the characteristics of chemical compositions, we can conclude that coal combustion and dust were the most important contributors to the local PM at the Lijiang site, while biomass burning and vehicle exhaust may also have contributed to some degree.

## Figures and Tables

**Figure 1 ijerph-17-09553-f001:**
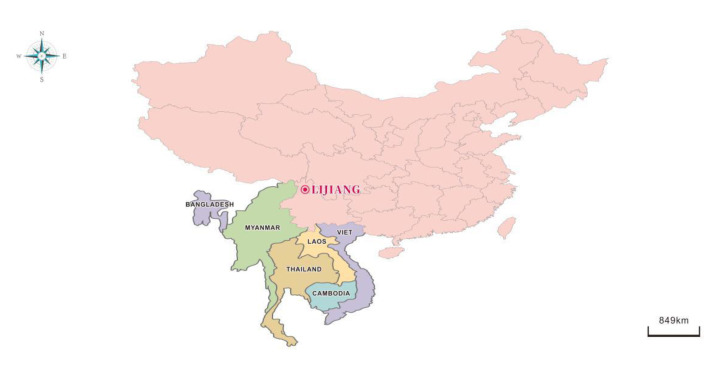
Map of sampling site.

**Figure 2 ijerph-17-09553-f002:**
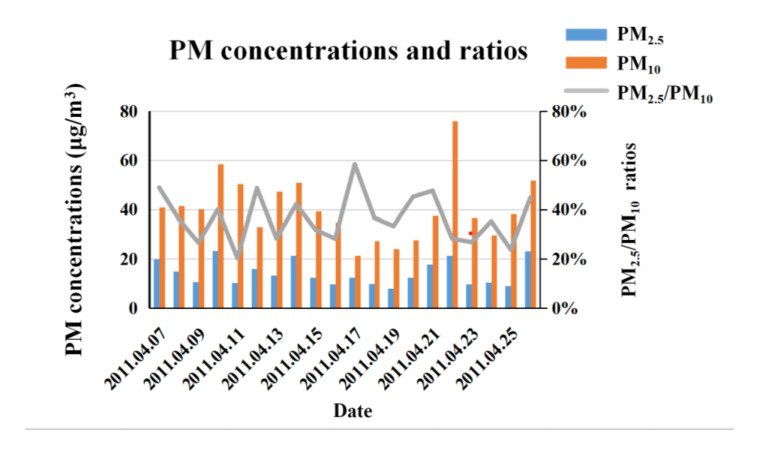
PM_2.5_ and PM_10_ mass concentration and size fraction.

**Figure 3 ijerph-17-09553-f003:**
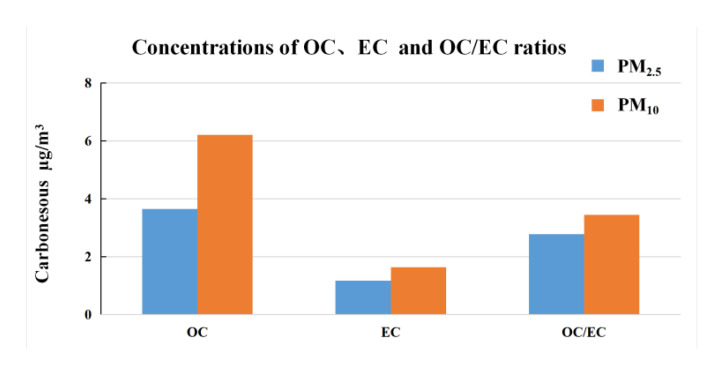
Monthly mean concentrations of organic carbon (OC) and elemental carbon (EC) and OC/EC ratio in PM_10_ and PM_2.5_.

**Figure 4 ijerph-17-09553-f004:**
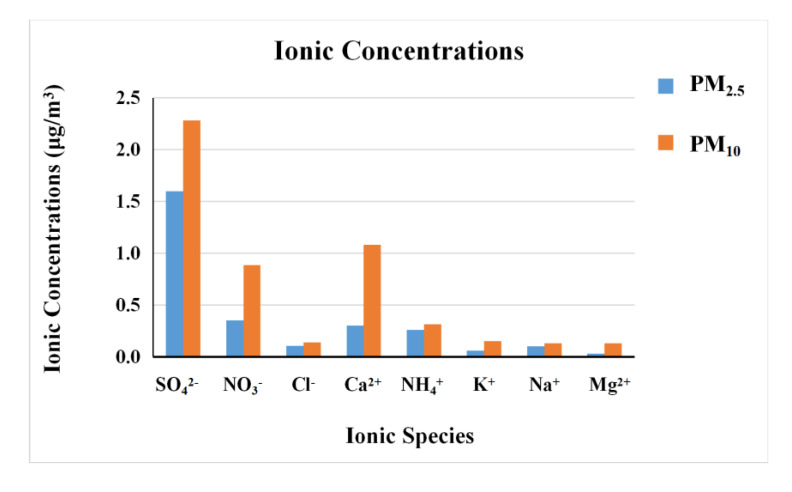
Mass concentrations of cation and anion in PM_10_ and PM_2.5._

**Figure 5 ijerph-17-09553-f005:**
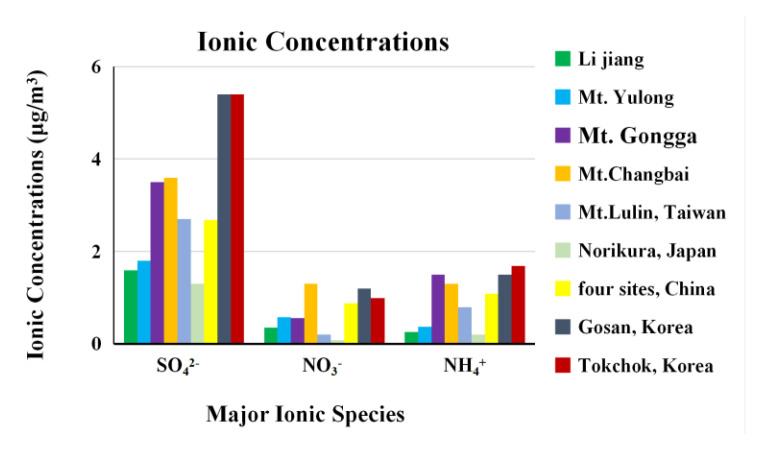
Major ionic concentrations in PM_2.5_ in different rural sites.

**Table 1 ijerph-17-09553-t001:** The detected compounds of PAHs and NPAHs in this study.

No.	Compound	Abbreviation	Rings
1	Fluoranthene	Flt	4
2	Pyrene	Pyr	4
3	Benz(a)anthracene	BaA	4
4	Chrysene	Chr	4
5	Benzo(b)flfluoranthene	BbF	5
6	Benzo(k)flfluoranthene	BkF	5
7	Benzo(a)pyrene	BaP	5
8	Dibenz(a,h)anthracene	DBA	5
9	Benzo(g,h,i)perylene	BghiPe	6
10	Indeno(1,2,3-cd)pyrene	IDP	6
11	1,6-Dinitropyrenes	1,6-DNP	
	1,8-Dinitropyrenes	1,8-DNP	
	1,3-Dinitropyrenes	1,3-DNP	
	9-Nitrophenanthrene	9-Nph	
	2-Nitroanthracene	2-NA	
	9-Nitroanthracene	**9-NA**	
	4-Nitropyrene	4-NP	
	3-Nitrofluoranthene	3-NFR	
	1-Nitropyrene	1-NP	
	1-Nitrofluoranthene	1-NFR	
	2-Nitrotriphenylene	**2-NTP**	
	6-Nitrochrysene	6-NC	
	7-Nitrobenz[a]anthracene	7-NBaA	
	3-Nitroperylene	3-Nper	
	6-Nitrobenzo[a]pyrene	6-NBaP	
	1-Nitroperylene	1-NPer	
	3-Nitrobenz[a]anthracene	3-NBA	
	2-nitrofluoranthene	2-NFr	

**Table 2 ijerph-17-09553-t002:** PM_10_ and PM_2.5_ concentrations and ratios at clean/rural sites (μg/m^3^).

Sites	Countries	Periods	PM_2.5_	PM_10_	PM_2.5_/PM_10_ (range)	Literature
Lijiang	China	2011, Spring	14.4	40.4	0.37(0.20–0.58)	This study
Changbaishan	China	2013	13	18	(0.55–0.84)	[24,25]
Pangquangou	China	2013	18	44	(0.17–0.74)	[24,25]
Shennongjia	China	2013	15	38	(0.24–0.75)	[24,25]
Nanling	China	2013	21	30	(0.54–0.82)	[24,25]
Gosan	Korea	2007–2008	17.2	28.4		[26]
Tokchok	Korea	1999–2000	18.7			[27]
K-puszta (PM_2_)	Hungary	2003, Summer	13.6	24	(0.57 ± 0.06)	[21]
K-puszta	Hungary	2006, Summer	17.4	25	(0.67 ± 0.08)	[28,29]
Illmitz	Austria	1998–2010	20	25		[30]
Langenbruegge	Germany	1998–2010	13	17		[30]
Payerne	Switzerland	1998–2010	17	20		[30]
Penausende	Spain	1998–2010	8	12		[30]

PM_2.5_/PM_10_ means the PM_2.5_ fraction in PM_10_, and with the range in brackets.

**Table 3 ijerph-17-09553-t003:** Daily and weekly mean concentrations of polycyclic aromatic hydrocarbons (PAHs) in atmospheric PM_10_ of Lijiang (ng/m^3^).

Abbreviation	6 Apr.	7 Apr.	8 Apr.	9 Apr.	10 Apr.	11 Apr.	12 Apr.	13 Apr.	Mean ± SD
Flt	2.9	2.9	3.2	5.9	1.9	1.6	1.3	2.0	2.7 ± 1.5
Pyr	1.9	1.9	1.9	3.3	1.6	1.1	0.8	1.4	1.7 ± 0.8
BaA	1.1	1.0	1.2	2.3	0.7	0.5	0.3	0.6	1.0 ± 0.6
Chr	1.8	1.7	2.2	3.4	1.1	0.9	0.6	1.0	1.6 ± 0.9
BbF	1.4	1.2	1.6	2.3	0.9	0.8	0.6	1.1	1.2 ± 0.5
BkF	0.8	0.7	0.8	1.3	0.5	0.4	0.3	0.5	0.7 ± 0.3
BaP	1.1	1.1	1.2	2.0	0.9	0.7	0.6	0.9	1.1 ± 0.4
DBA	0.03	0.02	0.04	0.08	0.03	0.03	0.01	0.02	0.03 ± 0.02
BghiPe	1.0	0.8	1.0	1.4	0.7	0.7	0.6	0.9	0.9 ± 0.3
IDP	1.1	1.0	1.3	2.0	0.7	0.6	0.5	0.8	1.0 ± 0.5
Total	13.0	12.2	14.4	23.8	9.1	7.5	5.7	9.3	11.9 ± 5.6

**Table 4 ijerph-17-09553-t004:** Diagnostic ratios of PAH isomers from different emission sources and atmospheric particles.

	IDP/(BghiPe + IDP)	BaA/(Chr + BaA)	Flt/(Flt + Pyr)
This study	0.53 ± 0.04	0.45 ± 0.01	0.61 ± 0.03
Biomass burning	0.48–0.58 [45,46,47,48]	>0.5 [49]	0.43–0.58 [45,46,48]
Coal combustion	>0.5 [47,48]	0.2–0.5 [47,49]	>0.5 [50]
Diesel vehicles	0.35–0.70 [47,48]	<0.2 [47]	0.6–0.7 [50]
Gasoline vehicles	<0.2 [47]	0.2–0.35 [49]	<0.5 [48,50]

**Table 5 ijerph-17-09553-t005:** Daily and weekly mean concentrations of nitrated polycyclic aromatic hydrocarbons (NPAHs) in atmospheric PM_10_ of Lijiang (pg/m^3^).

Abbreviation	6 Apr.	7 Apr.	8 Apr.	9 Apr.	10 Apr.	11 Apr.	12 Apr.	13 Apr.	Mean ± SD
1,6-DNP	0.14	0.12	0.15	0.12	0.10	0.11	0.08	0.09	0.11 ± 0.02
1,8-DNP	0.20	0.24	0.32	0.26	0.20	0.22	0.17	0.17	0.22 ± 0.05
1,3-DNP	0.24	0.14	0.18	0.15	0.12	0.12	0.09	0.11	0.14 ± 0.05
9-Nph	ND	7.37	8.48	12.28	4.24	ND	ND	ND	8.1 ± 3.3
2-NA	0.63	0.69	0.83	0.92	0.45	0.42	0.36	0.47	0.59 ± 0.20
9-NA	92	62	89	115	50	38	15	69	66 ± 32
4-NP	0.54	0.67	0.77	0.79	0.47	0.49	0.52	0.64	0.61 ± 0.12
3-NFR	0.74	0.87	1.03	0.76	0.55	0.57	0.40	0.49	0.68 ± 0.21
1-NP	12.12	12.61	13.35	9.64	9.64	11.13	8.65	8.90	10.8 ± 1.8
1-NFR	7.18	17.11	22.81	8.24	3.80	10.98	17.11	4.01	11.4 ± 6.9
2-NTP	386	307	302	328	171	137	38	168	230 ± 118
6-NC	21	22	29	34	16	20	16	21	22 ± 6.0
7-NBaA	15.58	9.57	10.11	9.02	5.74	6.29	2.51	5.47	8.0 ± 4.0
3-Nper	0.89	0.95	1.31	0.89	0.80	0.77	0.77	0.80	0.90 ± 0.18
6-NBaP	3.87	3.57	4.16	2.82	2.53	2.68	2.38	2.35	3.04 ± 0.71
1-NPer	1.78	1.13	1.19	1.52	1.22	0.83	0.80	1.07	1.19 ± 0.33
3-NBA	ND *	ND	ND	ND	ND	ND	ND	ND	
2-NFr	ND *	ND	ND	ND	ND	ND	ND	ND	
Total	450	376	385	396	213	191	87	212	289 ± 129

*: Detection limit is defined as 3 times the ratio of signal vs. noise, the detection limits of 3-NBA and 2-NF are 400 fmol and 1000 fmol, respectively.

**Table 6 ijerph-17-09553-t006:** Concentrations of elements in PM_2.5_ of different sites (μg/m^3^).

Year, Location, Ref.	Ca	Mg	Fe	K	Mn	Zn	Cu	Cr	As
2011, Lijiang, this study	0.59	0.18	0.16	0.083	0.0031	0.017	0.0083	0.0080	0.00018
2013, 4 NAB sites [24]	0.13	0.10	0.050	0.16	0.0080	0.020	0.0035	0.0020	0.0027
2006, Mt. Dinghu [65]	0.83	0.16	0.57	1.36	0.033	0.43	0.060	UD	0.031
2008, Xinglong * [66]	2.5	0.54	0.89	0.85	0.029	0.14	0.043	0.079	0.013
2013, Kunming [67]	ND	ND	ND	ND	0.16	0.33	0.078	0.030	0.03
2012, Beijing [68]	ND	ND	ND	ND	0.082	0.23	0.040	0.0078	0.012

* PM_2.1_ was studied in Xinglong instead of PM_2.5_. ND, not to be determined; NAB, national atmospheric background.

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
