# Peer review of "Chemical Characteristics of Atmospheric PM10 and PM2.5 at a Rural Site of Lijiang City, China"

_ijerph, 2020, doi:10.3390/ijerph17249553_

Round 1

Reviewer 1 Report

The manuscript titled: “Chemical Characteristics of Atmospheric PM10 and PM2.5 at a Rural Site of Lijiang City, China” presents a thorough assessment of several pollutants in a rural area of China, aiming at evaluate the major contribution of outdoor air pollution sources in that area. In general,  the manuscript is clear although some revisions are suggested in order to improve the presentation of study findings:

-  please carefully check and revise the text that would benefit from a thoughtful English grammar revision since some typos are present.

- in the Abstract, please spell out all abbreviations the first time are used (e.g. OC and EC, and PAH and NPAH). Similarly, please check throughout the manuscript that all abbreviations have been spell out the first time are used as well as they are used in tables and figures (especially Tables 3 and 4). This will increase the readability of the manuscript for those who are not familial with all pollutants assessed in the research.

- Abstract in general is focused on results of the study, but also background information and objectives of the research should be clearly stated at the beginning, while they came out only at the end when presenting major contributors of air pollution. I recommend to add at least the aim of the study in order to point out why several pollutants and elements have been assessed.

- sentence at L183-185 better belongs to Methods.

- paragraph of meteorological effects may be also presented earlier in the manuscript in order to confirm that particulate matter assessed in the study might be related to air mass transportation from South Asia, which is one of the background assumptions of the study.

- in Figure 2, please clarify that the gray line is the PM2.5/MP10 ratio.

Author Response

Dear reviewers,

The authors appreciate the reviewers for their useful and directly suggestions. Your valuable advice would illuminate our future scientific research work, and we will try our best to find the high-lights and significance of each projects. We have tried to write in a proper and clear way in English; but the grammar and the thinking logistic between Chinese and English are completely different. We will resort to a professional translation company to modify the English manuscript after the first review. The explanations and corrections according to the comments are as follows. We've given a response for each question/comments one by one.

The observation in this manuscript was aim to explore the effect on Southwest China of biomass burning of long range transport from Indo-China Peninsula region during the period of the southeast monsoon prevailed. Southwest China is mostly affected by the southeast monsoon in every April. Thus, these 20 samples could represent the character of this period. That is the reason why we only collected relative few filter samples within 20 days.

It is hard to recognize the contribution weights from different pollution sources because only 20 samples were too less to be used for the quantitative calculation by PMF or CMB. Moreover, there are a lot of potential sources of polluted metals of atmospheric PM2.5 in China, so, we do not like to jump into a conclusion without very conformed proofs. As you know, factories have been set up anywhere even in country side in China.

Fortunately, these samples were analyzed comprehensively, even including PAHs and N-PAHs, from which we could draw a cautious conclusion that the local atmospheric PM2.5 in Lijiang sampling site were really influenced by the biomass burning of long range transport from Indo-China Peninsula region.

The manuscript has been modified with the revision mode, so any change was recorded.

Best regards

Wan WANG

Reviewer 2 Report

Summary- The paper presents 20 measurements of fairly detailed composition analysis of fine and coarse particulate matter collected at a high-altitude mountainous site in southern China during April 2011. The measurements are extensively compared with literature searches of similar measurements made throughout the world. The main conclusion is that coal, biomass burning, vehicle exhaust and soil dust had "major" contributions to the measured particulates. This conclusion seems obvious, as they've listed all standardly accepted sources for particulates, so there is nothing really new here. The paper is filled with unacceptable numbers of minor English grammar problems where awkward wording, incorrect verb tense, subject references etc. often confuse what the authors are trying to communicate. The paper must be proofed for these minor errors throughout the text. The measurements are interesting, and should be available to the broader scientific community, but it seems that just 20 measurements are much too few for a "critical mass" as far as advancing the overall scientific significance of the results presented. If they had a years worth of measurements, the paper would provide much more valuable insights, but if this is all they have...? The paper could be published with lots of revisions suggested below, or rejected on the grounds that there is not too much raw new measurements presented. Suppose the authors only had 3 days of measurements, would that be adequate? Having only 20 individual days of measurements seems near the lower limit of a "least publishable amount" of measurements.

1) abstract: are the sulfate/nitrate ratios reported mass, mole, or equivalent ratios? State this explicitly.

2) abstract: how is 'enrichment' defined? relative to sea water? soil? See note below about how this enrichment is not too useful. The conclusion that the ratios of various elements in atmospheric particles differs from the ratios in local soils only says that particles are probably not local soils, and says nothing about anthropogenic pollution contributions.

3) line 54-56. Very confusing English wording. Monthly average reaches 20 micrograms/m3 contribution rate 30%, as twice as from local. Do they mean that 20 ug/m3 is 30% of the total loading? and that local contributions are about 15% contribution?

4) line 62 again confusing wording: Indochinese Hg is "11.4 Mg/yr (units in text are vague) annual equivalent to 40%". 40% of What??

5) line 75-76: another example of confusing/meaningless wording: "Present study is located ... on the transmission channel of spring monsoon". What does this mean?

6) Fig 2: The right vertical axis is labeled "rartis". Do they mean ratio?

7) line 135: The authors note that concentrations measured are close to what have been deemed background at a few other sites across China, and therefore conclude that there are no local influences. This is not scientifically justified. It is entirely possible that local factors are contributing. Just remove this paragraph. Its OK to compare these measurements with other measurements, and leave it at that.

8) in all of the concentrations presented (figure captions, tables) the TIME AVERAGING period for the measurements must be clearly stated, and only measurements with the same time averaging periods should be compared. For example, Daily average (24-hr) measurements should not be compared with monthly averages. If the authors are comparing the average over their 20 days of measurements, and the Korean numbers are Monthly averages, these two different time averaging periods must be explicitly stated. I would suggest that all the PM concentrations noted in Table 1-2 be noted by their numerical value, followed by the averaging period: For example 14.1 (annual) or 12.2 (3-month). The authors note there are some measurements of "warm" and "dry" periods. How long were these warm and dry periods? Months? weeks? Seasonal (3-month)?

9) for Fig 3, in the caption, the time average of the measurements must be stated: "20-day average OC, SOC and EC concentrations" (or whatever the averaging time is).

10) Line 167, the authors note the mean, then compare with concentration ranges. Again the TIME AVERAGE period of all concentrations presented MUST BE STATED. E. g. "The 20-day mean was 3.6 ug/m3, and their DAILY-AVERAGE concentrations ranged from 1.7-7.9"

11) Fig 3: I am confused about SOC. Apparently SOC is ONLY computed from OC and EC? Using the minimum of the OC/EC measured during the 20-day measurement period? I am not familiar with the reasoning being this equation on line 184, and there is no reference to how it was derived, and I doubt that the minimum over a single 20-day period is applicable to all measurements during the period? Get rid of all references to SOC if SOC is only calculated from EC and OC. Just show raw measurements. If SOC is calculated from raw measurements, delete all references to SOC.

12) Line 208 notes that 11 kinds of PAHs are measured. Table 2 lists only 10 kinds. What was the 11th PAH, and why isn't it listed in Table 2?

13) line 209 notes total PAHas 11.9, yet table 2 shows total as 10.2. I assume the difference is because there is a missing PAH in Table 2? Please clarify.

14) the three letter codes for the PAHs listed should somewhere be defined, maybe in a table footnote? or just type out the whole name in the table and don't use the shortened 3-letter code.

15) Table 4 has the same problem as noted for Table 3 above: 16 nPAHs are listed, but the text notes that "18 kinds" are measured. Why aren't the other two shown in the table? At least in Table 4 the total is consistent between the table and the text describing the table. Please clarify why there is a discrepancy (16 vs. 18)

16) line 297 two percentages are noted 13% and 23%. I assume these percentages of the total mass concentrations? Please state this.

17) line 307. The authors note the percentages of various cations. I assume these are all MASS percentages? please explicitly state this. Many ion measurements are reported as equivalents, and mole ratios have more usefulness to many researchers, so this should be explicitly clarified.

18) line 315 discusses SECONDARY inorganic ions. What are 2nd inorganic ions? Please define. Are these simply sulfate and nitrate? Is ammonium considered secondary? If so, why? I would remove this reference to secondary. Simply state sulfate, nitrate, & ammonia ions ...

19) line 365: the authors discuss their measurements in terms of "enrichment factors", relative to some generic crustal component. I don't think this is at all scientifically relevant. Surely different crustal rocks or soils have widely varying ratios of elements relative to Fe, so what particular crustal element profile are they using here and why? Simply noting that airborne particles have different chemical compositions from soil is obvious and by itself is NOT an indicator that anthropogenic pollution dominates. As noted above, the only thing one can conclude from differences from soil is that measured particles probably weren't soil-derived.

20) Line 384 describes backward air trajectory analysis. Not much information is provided here that is useful for interpretation. Simply showing a SURFACE wind rose for the measurement period would be sufficient. air arriving 3000m above the ground (at what time of day?) probably doesn't say anything at all about ground level trajectories, and furthermore, in mountainous areas like this study, lower troposphere trajectories are probably next to meaningless unless there are fairly high resolution MEASUREMENTS of wind speed and direction through out the valleys and mountains performed at hourly intervals. If trajectories are presented, please show 3-5 trajectories from 4-5 levels starting at the ground and at go no higher than the typical planetary boundary layer depth of 1-2 km, and do these trajectories at least 3-4 times per day. Either that, or simply show a plot of wind speed and direction at 1-2 sites near the measurement site.

21) After deleting the trajectory analysis, the authors should simply show temperature, winds, pressure, rainfall during the measurement period.

Author Response

(The authors gave the same response as above.)

Round 2

Reviewer 2 Report

Summary- During this second review, I found so may flaws still in the paper that I found it frustrating to read within the time frame of the review deadlines for this Journal. At the last point in my comments below, I gave up after seeing such sloppy typos where the text discussion of Table 3 does not agree with the numbers presented in the table (the means match, but the variation doesn't). I give up. This paper should not be published. With only 20 measurements, It does not contain enough new measurements worthy of an entire publication.

Some suggestions the authors have to address in any attempt to publish this elsewhere:

1) The authors have removed concentrations that measured as zero from their tables. A measurement below detection limit is a valid measurement, and these below detection limits should just be stated, rather than eliminated from the table altogether. Simply state "<1 ng/m3" or whatever the detection limit is. Please do not delete valid measurements.

2) I now understand their sentence about IndoChinese mercury emissions. It would be more clear to simply state: "TOTAL mercury emissions from Indochina are estimated to be 28.5 Mg/yr, and 40% of that total is from biomass burning"

3) The authors are incorrect in identifying NH4+ as originating from chemical reactions. Ammonia gas is emitted, and partitions (dissolves) into hazy particles. There is no chemical reaction involved that chemically converts the ammonium species. Dissolving in particles or haze solutions is not a "reaction". There is no need for calling anything "primary" or "secondary". If by secondary the authors mean sulfate, nitrate, and ammonium, simply state that. Authors here ignored my pointing this out 1st review)

4) The author's defense of using trajectories from 3000m above the ground is incorrect, As mentioned in my original review, unless the authors can show trajectories from 2-3 levels (base, middle and top of planetary boundary layer) and calculate trajectories at these levels 3-4 times per day, this section discussing trajectories is overly complicated and meaningless. In their comments, the authors claim that trajectories from below 2400m might be stopped by mountains is completely true! Mountains block and redirect movement of air, and the incorrect results shown here from 3000m trajectories suggest that the mountains DON'T block air. This section on trajectories must be improved as noted above, or deleted altogether.

I suggested showing wind measurements from the collection site, and the authors apparently do not have measurements. Thats OK. Then simply show the winds that HYSPLIT uses averaged for the month near the surface. Either that or 2-3 times/day back trajectories for 1-2 hours ONLY at the lowest level of the HYSPLIT model that claims to represent winds over their measurement site. Say something like: Hysplit winds in the lowest 100m above the surface averaged 10 m/s out of the southwest throughout the measurement period, or whatever the average is.

However, its not even necessary to quantitatively show transport from India or Indochina. A simple reference to a paper showing Monsoon circulations at the surface would suffice as long as those references show movement some of the time from anywhere in the southern quadrant to the measurement site. I recommend removing all references to this flawed trajectory analysis, and simply note that winds come from the south sometimes during April.

5) Line 82: the station location is listed has having a latitude of 26deg 86'. There are only 60' (' - means minute or arc) in a degree. is this 86' a typo?

6) Line 121: The authors mention that temperatures varied from 1-5C. Is this correct? This sounds like much too small of a variation. Day/night temperatures at most continental locations fluctuate by 10-20C. The same applies to the wind range 1-6 m/s. There were no calm periods throughout an entire month? No gusts? no fronts?

7) At several locations the authors have not stated the time averaging period for measurements numbers presented, e. g. Line 140: please state that DAILY AVERAGE concentrations range from 21-76. Line 176: the DAILY concentrations ranged from

8) Line 195: The authors present SOC that is simply calculated from measured EC and OC. Despite their defense of this method, and a reference for its use, I disagree that this presents anything scientific. It should be eliminated as unsound science inferences, and most likely NOT generally applicable everywhere. The "minimum" ratio during some measurement period will vary widely if one only uses a weeks worth of measurements rather than a month or year's worth. What if raw measurements were hourly? would one use the minimum HOURLY ratio?

9) Line 235 Authors note that nenzo[b]fluoranthene was found at night to be 2.9 ng/m3 +/- 5.4. Please state that these are HOURLY concentrations? furthermore, with the range higher than the mean by a factor of two, the authors should just state the range of the measurements.

10) line 236: Faulty reasoning: Authors note that Mt. Wuzhishan has annual mean of 0.42, and Lijiang has 2.7, then conclude "Lijiang has rather great biomass burning". This implies that simply because one measurement is greater than another, that biomass burning is "rather great"?

11) line 240: again authors do not state the time averaging period for of the ranges of BaP concentrations shown. Hourly? Daily? Weekly?

12) Description of Table 3: Three columns of ratios are presented. The ranges of these ratios are discussed in the following paragraph, and the VARIABILITY of these numbers are grossly different in the text description of these rations from what is presented in the table

Column 1 lists 0.53+/-0.04 text line 264 says 0.53+/- 0.66
Column 2 lists 0.45+/-0.01 text line 266 says 0.45+/- 0.46
Column 3 lists 0.61+/-0.03 text line 268 says 0.61+/- 0.66

While the means match perfectly, the variability is grossly different, and line 262 notes that the ratios are "quite constant'. Why the discrepancy between the variability listed in the table and referenced in the text??

Furthermore, the reasoning in lines 260-270 is unsound! Apparently one ratio (IDP/BghiPe+IDP) suggests biomass (but also Coal & diesel). The other two ratios are clearly outside the range of the biomass ratios. Yet the authors conclude that "biomass burning has an impact". I don't doubt that conclusion, but when two ratios show that it CANNOT be biomass, while ALL THREE Ratios are in the range of coal & diesel, clearly coal and diesel are much more important. The concluding sentence MUST be that in all likelihood coal and diesel are probably the most important source of the particles measured, and NOT biomass. I don't doubt that biomass is influencing these measurements, but these ratios DO NOT demonstrate that at all!

During this second review, I found so may flaws like this last flaw (typo/inconsistent table & text) still in the paper that I found it frustrating to read within the time frame of the review deadlines for this Journal. At this point in the manuscript, I gave up after seeing such sloppy typos where a table cannot agree with the text discussion of the table. I give up. This paper should not be published. It dopes not contain enough measurements worthy of disseminating to the scientific community

Author Response

Dear reviewer,

Thank you very much for your review.

The point-to-point reply in my attachment.

Sincerely,

Xurui Li.
